

# Vertical Velocity Variance Measurements from Wind Profiling Radars

Katherine McCaffrey[1,2], Laura Bianco[1,2], Paul Johnston[1,2], and James M. Wilczak[2]

[1]University of Colorado, Cooperative Institute for Research in Environmental Sciences at the NOAA Earth System Research Laboratory, Physical Sciences Division, 325 Broadway, Boulder, CO 80305-3337
[2]NOAA Earth System Research Laboratory, Physical Sciences Division, 325 Broadway, Boulder, CO 80305-3337

*Correspondence to:* Katherine McCaffrey (katherine.mccaffrey@noaa.gov)

**Abstract.** Observations of turbulence in the planetary boundary layer are critical for developing and evaluating boundary layer parameterizations in mesoscale numerical weather prediction models. These observations, however, are expensive, and rarely profile the entire boundary layer. Using optimized configurations for 449 MHz and 915 MHz wind profiling radars during the eXperimental

Planetary boundary layer Instrumentation Assessment, improvements have been made to the historical methods of measuring vertical velocity variance through the time series of vertical velocity, as well as the Doppler spectral width. Using six heights of sonic anemometers mounted on a 300-m tower, correlations of up to $R^2 = 0.74$ are seen in measurements of the large-scale variances from the radar time series, and $R^2 = 0.79$ in measurements of small-scale variance from radar spectral

widths. The total variance, measured as the sum of the small- and large-scales agrees well with sonic anemometers, with $R^2 = 0.79$. Correlation is higher in daytime, convective boundary layers than nighttime, stable conditions when turbulence levels are smaller. With the good agreement with the *in situ* measurements, highly-resolved profiles up to 2 km can be accurately observed from the 449 MHz radar, and 1 km from the 915 MHz radar. This optimized configuration will provide unique

observations for the verification and improvement to boundary layer parameterizations in mesoscale models.

## 1 Introduction

Observations of turbulence quantities in the planetary boundary layer (PBL) are crucial for many applications, and in particular, can be extremely informative for developing and evaluating param-

20 eterizations in numerical weather prediction models of the small scales that cannot yet be resolved. However, turbulence measurements are predominantly relegated to high-frequency *in situ* observing instrumentation such as sonic anemometers, limited in their spatial coverage, or are taken by expensive aircraft platforms. Lidar remote sensing instrumentation have demonstrated some potential



for measuring profiles of turbulence (Eberhard et al., 1989; Frehlich, 1997; O'Connor et al., 2010),
but this technology has more commonly focused on mean wind measurements (Menzies and Hard-
esty, 1989; Grund et al., 2001; Lundquist et al., 2016). Similarly, wind profiling radars (WPRs) have
been shown to have capabilities of measuring turbulence, from information contained in the Doppler
spectral width of the vertical velocity (Hocking, 1985; Reid, 1987; Angevine et al., 1994; Nastrom
and Eaton, 1997), but the adoption of these techniques into routine use has not occurred because
of the lack of precision and inability to measure the smallest turbulence values observed by sonic
anemometers.

In the full energy spectrum, contributions to the total variance come from large to small scales, the
separation of which is determined by different instruments' measurement frequencies and volume
sizes. In general, the total variance can be assumed to be the sum of the large and small scales
(Angevine et al., 1994):

$$\text{Total Variance} = \text{Large Scale Variance} + \text{Small Scale Variance} \qquad (1)$$

For a WPR, the contribution from the large scales can be obtained using the times series of the re-
solved vertical velocity, and the contribution from unresolved scales that are smaller than the pulse
volume can be indirectly estimated through the Doppler spectral width of the vertical velocity. How-
ever, conventional WPR configurations are usually not adequate for measuring very small turbulence
scales, because accurate measurement of the spectral width contributions due solely to turbulence is
not trivial, as other factors, such as the beam-width of the radar antenna, and horizontal and vertical
shear of the horizontal winds inside the volume of measurements, act to broaden the spectral widths.
Nevertheless, previous studies have used the Doppler spectral width of vertical velocity with partial
success, for calculation of eddy dissipation rates. On the other hand, the typical temporal resolution
of time series of first-moment velocities limits the usage of WPRs for direct measurements of the
large scale contribution to the total variance. Angevine et al. (1994) used a 915 MHz WPR (Ecklund
et al., 1988), to measure vertical velocity variances over both large and small scales by combin-
ing the contributions from the time series and spectral widths of the vertical velocity, respectively.
However, the purpose of that study was not the optimization of the radar for variance observations,
but the measurement of the vertical heat flux. Furthermore, due to the coarser spectral and temporal
resolution of that system, the variances were analyzed over 2-hour periods, and relied on the vertical
component of velocity from the oblique beams to increase the resolution for large-scale variance
measurements.

This study aims to accurately measure the total variance, as well as the individual contributions
from large and small scales, with optimized WPR configurations and post-processing procedures.
Here, we use two WPRs operating in this optimally-defined "turbulence mode" during the eXper-
imental PBL Instrumentation Assessment, XPIA, to observe profiles of vertical velocity variance,
obtaining information on the large scale from the time series of vertical velocity, and information on
the small scales from the Doppler spectral widths of the vertical velocity. The confirmation of the





ability of the optimized WPR set-up and post-processing methods to measure accurate variances at different scales allows the usage of this remote-sensing instrument for a larger variety of applications.

## 2 Observations

All observations used for this study were gathered at the Boulder Atmospheric Observatory (BAO), located in Erie, Colorado, and operated by the National Oceanic and Atmospheric Administration's Earth Systems Research Laboratory (Kaimal and Gaynor, 1983). The site is in gently rolling terrain, about 30 km north of Denver and 20 km east of the foothills of the Front Range of the Rocky Mountains. The centerpiece of the site is the 300-meter meteorological tower, routinely instrumented at
10, 100 and 300-m with temperature, humidity and velocity sensors. During the Spring of 2015, XPIA ran from 1 March to 1 June 2015, with the goal of assessing the ability of remote-sensing instruments, including profiling and scanning lidars, microwave radiometers, and profiling and scanning radars to observe the PBL (Lundquist et al., 2016). Two wind profiling radars (915 MHz and 449 MHz) were operating as part of the project, set up specifically to measure turbulence. For XPIA,
the number of instrumented heights of the 300-m BAO tower was increased, with pairs of sonic anemometers on opposite sides of the tower at 6 heights. These tower measurements serve as the *in situ* observations against which the remote sensing observations from the WPRs will be compared. Both sonic anemometer and WPR variance quantities are calculated over 30-minute sampling periods.

### 2.1 Sonic Anemometers

During XPIA, the BAO tower was equipped with 12 Campbell Scientific CSAT3 sonic anemometers (commonly referred to simply as "sonics"), two at each height every 50 m from 50 to 300-m on southeast- and northwest-facing booms, at $154°$ and $334°$ from north, respectively. All sonic anemometers measured at 20 Hz, with a measurement resolution (offset error) of $0.1\,\mathrm{cm\,s^{-1}}$ ($8\,\mathrm{cm\,s^{-1}}$)
in the horizontal and $0.05\,\mathrm{cm\,s^{-1}}$ ($4\,\mathrm{cm\,s^{-1}}$) in the vertical. The northwest sonic anemometers were functioning throughout the experiment, and the southeast sonic anemometers were available as follows: 100 m began running on 1 March 2015; 50, 150, 200, and 250 m began on 3 March; and 300-m began on 7 March. The heights of the sonic anemometers overlapped with six of the 915 MHz profiler's range gates, as well as the bottom four 449 MHz gates, from 150 m and above (see
section 2.2 for the WPRs' specifications). The pairs of sonic anemometers were averaged together, except when one boom was in the tower wake, i.e., when the 1-minute mean winds were blowing through the triangular tower from $288 - 28°$ and $104 - 189°$ (from N), as determined by McCaffrey et al. (2016, in revision). Figure 1 is the wind rose from the northwest sonic anemometer at 200 m. The winds coming from the direction of the tower have been removed. Sonic data, sampled at



20 Hz, were also excluded if the sonic signal amplitude was too low or high, if the signal lock was
poor, or if the difference in the speed of sound between the three non-orthogonal axes was too high
(internal instrument quality control). The sonic anemometers recorded three-directional velocities,
aligned with **u** directed into the boom, and **v** 90-degrees to the left. A planar tilt correction algorithm
developed by Wilczak et al. (2001) was applied to the data to first remove any possible vertical tilt

of the instrument (which was $< 2°$ in all cases), and to realign the velocities so that $\bar{\mathbf{u}}$ is coordinated
in the 30-minute mean wind direction and $\bar{\mathbf{v}} = 0 \text{ m s}^{-1}$. These aligned velocities were then used in
all calculations of vertical velocity variance.

### 2.2 Wind Profiling Radars

The two wind profiling radars used during XPIA were a 449 MHz and a 915 MHz WPR, both located

near the BAO visitor's center (the 915 MHz to the west, the 449 MHz just to the south), about 600 m
to the southwest of the 300-m tower. The profilers collected data from 1 March until 30 April 2015
in a rotation of three modes each hour: for the first 25 minutes of each hour in "normal acquisition
mode," with collection of Doppler spectra for consensus winds from 3 beams (one vertical and two
oblique); for 30 minutes in "turbulence mode," with collection of time series of backscatter intensity

from only the vertical-pointing beam; and for the last 5 minutes of each hour in Radio Acoustic
Sounding System (RASS) mode. Backscatter intensity time series and Doppler spectra files were
post-processed to obtain raw data files containing radial velocity, spectral width, and signal-to-noise
ratio (SNR) for analysis.
    The radars measure the backscatter intensity of the atmosphere in quasi-cylindrical volumes of

length, $\Delta R$, and with a diameter that increases with distance from the radar. The backscatter time
series is then converted into a Doppler spectrum of velocities, $S(v)$, through a fast-Fourier transform
(FFT). The distribution of velocities observed in the volume determines the power ($0^{th}$ moment),
mean velocity ($1^{st}$ moment), and variance or width ($2^{nd}$ moment), of the Doppler spectrum. The
basic method of calculating the moments (standard or single peak-processing, SPP) finds the velocity

with the largest power at each height, then gathers the velocities, $v_1$ and $v_2$, on either side of the peak
with power greater than a threshold, typically the maximum noise level (Hildebrand and Sekhon,
1974), as the bounds of the integral used to calculate the moments as follows:

$$0^{th} \text{ moment} \quad = \quad P = \int_{v_1}^{v_2} S(v)dv \tag{2}$$

$$1^{st} \text{ moment} \quad = \quad \langle v \rangle = \frac{\int_{v_1}^{v_2} vS(v)dv}{P} \tag{3}$$

$$2^{nd} \text{ moment} \quad = \quad \sigma^2 = \frac{\int_{v_1}^{v_2} (v - \langle v \rangle)^2 S(v)dv}{P}. \tag{4}$$

The $2^{nd}$ moment, $\sigma^2$, is output as the spectral width, $\delta = 2\sigma$.



The length of time between each measurement (dwell time, $\Delta t$) is dependent on the product of several radar parameters including the inter-pulse period ($IPP$), the number of coherent integrations ($NCOH$), the number of points used in the fast Fourier transform ($NFFT$), and the number of
spectral averages ($NSPEC$):

$$\Delta t = [IPP][NCOH][NFFT][NSPEC]. \qquad (5)$$

The general post-processing methods for Doppler spectra include a routine to remove the contamination from non-atmospheric signals in the spectra, and then use a peak-processing algorithm to determine the first two moments (radial wind speed and spectral width). It is optional to perform a
number of spectral averages ($NSPEC$) in the post-processing procedure, resulting in lengthened dwell times. The impact generated by using a different number of spectral averages will be included in the analysis of variance measurements (Sect. 5).

     In the calculation of the Doppler spectrum from the time series of backscatter intensity, wavelet and Gabor post-processing methods are commonly used to filter contamination from birds, radio-
frequency interference, ground clutter, and other non-atmospheric signals. The wavelet algorithm acts on the time series of backscatter intensity to reduce the clutter from non-atmospheric frequency signals, and removes them before the FFT is computed (Jordan et al., 1997). Similarly, the Gabor filtering method also works on the time series to identify and remove non-stationary signals from birds and other point targets (Lehmann, 2012). A ground-clutter removal algorithm is also applied,
which removes any spectral peaks centered around $0 \text{ m s}^{-1}$. These processes provide significantly cleaner spectra and have been confirmed to improve estimates of the first moment (Bianco et al., 2013).

     Common peak-processing methods include the standard method described above (SPP), as well as the multiple peak-processing (MPP) method of Griesser and Richner (1998). This algorithm iden-
tifies the three largest peaks in the spectrum at each height of measurement, then uses continuity in time and space (vertical profiles) to identify the most-likely true peak. MPP was not used in this study because, though it has been shown to calculate more precise mean winds for typical radar setups (Gaffard et al., 2006), the high spectral resolution used in turbulence mode is incompatible with MPP, often identifying multiple peaks within one true peak, leading to greatly under-estimated
spectral widths.

     When using SPP, the threshold that determines the spectral width can be set to either the maximum or mean noise level of the spectrum. The common choice is to use the maximum noise level since it is the most conservative for removing noise, providing a better estimation of the first moment of the spectrum, and therefore this threshold was used for all first-moment calculations. However,
the choice of the maximum noise level can cause the spectral width to be underestimated. The mean noise level in these cases allows the measured spectral widths to be broader. Figure 2 exemplifies this, with a theoretical Gaussian signal plus added noise, with the mean and maximum noise levels shown with the dashed and dotted horizontal lines, respectively. The intersections between the Doppler



spectrum and the maximum noise level (dotted line) will occur at narrower velocity values than the
intersection with the mean noise level (dashed line). As a consequence, the use of the maximum
noise level will generate smaller spectral widths than those obtained using the mean noise level.
Therefore, we decided to use the mean noise level with SPP for measurements of Doppler spectral
widths.

Conversely, if the noise power contained in the Doppler spectra is too high (SNR is too low),
identification of the correct atmospheric peak may be prevented, or the peak may be falsely narrowed
(imagine moving the horizontal noise lines in Fig. 2 up). Using the method of Riddle et al. (2012),
a minimum threshold was applied to determine the usability for measuring the mean velocity of the
spectra based on SNR, $NFFT$, and $NSPEC$:

$$SNR_{min} = 10log\left[\frac{25\left(NSPEC - 2.3125 + \frac{170}{NFFT}\right)^{1/2}}{NFFT \times NSPEC}\right]. \tag{6}$$

This threshold was applied to each individual spectrum to determine if the first and second moments
are discernible through the noise. A discussion of the accuracy of width measurements based on
SNR can be found in the appendix.

During XPIA, the raw time series of backscatter intensity were collected in order for all post-
processing steps to be tested and optimized. The turbulence mode was configured with the goal of
capturing the fullest range of scales in the energy spectrum by increasing the number of dwells in
each 30-minute interval, and by maximizing the spectral resolution to capture the most accurate
spectral widths. This is accomplished by both minimizing $\Delta t$, while maximizing $NFFT$. Figure 3a
shows an example spectrum that has spectral resolution that cannot accurately capture the Doppler
width, despite the mean velocity being accurate. On the other hand, Fig. 3b shows how, with a differ-
ent set-up (more FFT points and fewer spectral averages on the same dwell), smaller spectral widths
can be captured. This example contains a ground-clutter peak at $0$ m s$^{-1}$, but the low resolution
cannot distinguish it from the true atmospheric peak, creating one broad peak. The higher spectral
resolution can distinguish the ground clutter, and therefore is able to it and accurately measure the
narrow width of the true peak. A spectral resolution on the order of $0.01$ m s$^{-1}$ was set, to guarantee
that spectral widths down to $0.1$ m s$^{-1}$ could be resolved using several points. Table 1 summarizes
the default parameters used in turbulence mode for calculating the Doppler spectra from the two
WPRs. The resulting dwell time for the 449 MHz WPR is 13 s, and 17 s for the 915 MHz WPR,
with $NSPEC = 1$ (spectral averaging can be performed in post-processing).

Since the 449 MHz WPR has a larger power-aperture product, and therefore a higher overall SNR,
the measured spectra are usually cleaner and the moments more accurate. For this reason our analysis
will first be performed on the data from the 449 MHz WPR, and later we will repeat it on the 915
MHz WPR to confirm the applicability to other radar systems.





## 3  Vertical Velocity Variance Calculations

When comparing vertical velocity variance from sonic anemometers, which measure velocity at very high frequency, and WPRs, which measure a Doppler spectrum at lower temporal resolution, multiple calculation methods must be applied for the resolved and unresolved scales. From the time series of the first moments of WPR Doppler spectra, the resolved, large-scale, 30-minute variance can be measured, $TS = \overline{w_r'^2}^{30}$, while the small-scale variance can be measured from the Doppler spectral width (second spectral moment), $SW = (\frac{1}{2}\delta)^2$. Equation 1 can be specified for the WPR, and the total WPR variance be computed as

$$\text{Total Variance}_{\text{WPR}} = TS + SW. \tag{7}$$

Since the WPR observes a volume, the finite beam-width of the radar antenna as well as the wind shear across the measurement volume will contribute to the broadening of the spectrum, generating larger spectral widths. Nastrom and Eaton (1997) have determined the shear and beam-broadening contributions, $\sigma_s^2$, on the observed width (in terms of spectral variance) to depend on both the mean wind transverse to the beam axis, $V_T$, as well as the antenna properties as

$$\sigma_s^2 = \frac{\nu^2}{3} V_T^2 cos^2\theta - \frac{2\nu^2}{3} sin^2\theta \left( V_T \frac{du}{dz} R_0 cos\theta \right) +$$
$$\frac{\nu^2}{24} \left(3 + cos4\theta - 4cos2\theta\right) \left(\frac{du}{dz}\right)^2 R_0^2 + \left(\frac{\nu^2}{3} cos4\theta + sin^2\theta cos^2\theta\right) \left(\frac{du}{dz}\right)^2 \frac{\Delta R^2}{12} \tag{8}$$

In the case of a vertical pointing beam ($\theta = 0^o$), this simplifies to

$$\sigma_s^2 = \frac{\nu^2}{3} \left( V_T^2 + \left(\frac{du}{dz}\right)^2 \frac{\Delta R^2}{12} \right) \tag{9}$$

where $\nu$ is the half-width to the half-power point in the antenna pattern, and $du/dz$ is the vertical mean wind shear. In our analysis, these effects have been subtracted from each dwell's observed spectral width, since the total variance is a sum of these independent contributions. In the cases when $\sigma_s^2$ is larger than the measured spectral width, the dwell was discarded. Though this may produce a high bias in the 30-minute WPR average, as seen by Dehghan et al. (2014), all other solutions (replacing the value with 0, allowing a negative spectral width, or substituting a small value) are not physically realistic, or are artificially created, causing statistical inaccuracies. Furthermore, fewer than 10% of the 449 MHz dwells had a situation of $\sigma_s^2$ larger that the measured spectral width (the 915 MHz is more impacted).

Appropriate averaging time scales must be applied to the sonic anemometer data for a direct comparison to WPR variances at small and large scale. For the resolved, large-scale variance, low-passed sonic anemometer variance (labeled "LP" on figures) is calculated from an averaged time series that matches the resolution of the WPR time series (dwell time, $\Delta t$). The variance is therefore calculated by first averaging the 20-Hz data to the dwell time of the WPR, $w_{\Delta t}$, and then computing the 30-minute variance as $LP = \overline{w_{\Delta t}'^2}^{30}$. The small-scale, high-passed variance from the sonic anemometers



(labeled "HP"), which contains all of the high-frequency information lost in the averaging in LP, is calculated computing the variance of the 20-Hz sonic data over the same dwell time of the WPR, as $HP = \overline{w'^2_{20Hz}}^{\Delta t}$. The high-frequency information contained in HP is thus equivalent to that of the spectral width of the WPR Doppler spectrum, and 30-minute averages of each can be compared.

The total variance from the sonic anemometers, with time-scale separation that matches the WPR resolution, is then obtained by (in the form of Eq. 1):

$$\text{Total Variance}_{\text{sonic}} = \text{LP} + \text{HP}. \tag{10}$$

Though instrument noise, $n$, is sometimes subtracted from the observed variance (Thomson et al., 2010), $n$ is negligible in relation to the velocity fluctuations, and will, therefore, be ignored in the

240 variance calculations herein. The agreement between the WPR and sonic anemometer measurements will be quantified using the mean difference or absolute error, normalized bias, and the coefficient of determination, $R^2$. Since the results are best presented on logarithmic scales, the $\log_{10}$ of all values is used for computing these variances.

The complete variance over 30-minutes of observations includes contributions from *all* time

scales, and thus the most accurate total variance can be obtained from the 20-Hz sonic anemometer data: $tot = \overline{w'^2_{20Hz}}^{30}$. It is therefore possible, from the sonic anemometer data, to determine if Eq. 10 is valid. If so, and if the WPR TS and sonic LP variances, and WPR SW and sonic HP variances are equal, then it can also be assumed that the sum of TS and SW variances will equal the total variance measured by the sonic anemometer. Each pair of sonic-WPR scales and their totals will be compared

in Sect. 4.

Each dwell collected by the 449 (915) MHz WPR spans about 13 (17) seconds, capturing only a short period of the atmosphere's motions. This leaves a large portion of the variance to the large scale, and the small scale variance by itself will not be representative of the turbulent flow, as it is missing a large portion of the energy spectrum. In the case of Doppler spectra from pre-determined radar

pulses, multiple dwells can be averaged to span a longer period of fluctuations (dwell time) resulting in more representative turbulence statistics. However, averaging over periods that are too long, and therefore non-stationary, will result in broadening the spectral peak due to a shifting mean velocity, rather than true fluctuations from turbulence. In this case, the SW variance will be unrealistically large, and the TS variance will lack resolution over the 30-minute period. Therefore, an analysis

was performed to determine the length of time, set by $NSPEC$, which produces the most accurate variances from the WPR (TS, SW, and Total Variance$_{\text{WPR}}$) compared to the *in situ* observations from the sonic anemometers.

## 4 Results from the 449 MHz WPR

Since the WPR is unable to resolve all scales of variance directly, its various contributions must

be compared to the equivalent contributions in the sonic anemometers' variance. This requires the





assumption, however, that the sum of the small- and large-scale contributions (sonic anemometers LP and HP variance and the equivalent WPR TS and SW contributions) is equal to the total variance over all scales, as calculated by the sonic anemometers. To confirm this, the sum of sonic LP and HP and Total Variance$_{\text{sonic}}$ are compared in Fig. 4. Though all data in this figure are from the sonic anemometers, the time scale of separation between LP and HP is determined by the un-averaged ($NSPEC = 1$) dwell time of the 449 MHz WPR of 13 s. The agreement is very good, with an $R^2$ value of $0.97$ and a mean difference of $-0.01$ m$^2$ s$^{-2}$.

With the confidence that the sum of sonic anemometers' LP and HP variance accurately calculates the full variance, the partitioned sonic's contributions can be compared to the WPR's. Figure 5 shows the comparisons between each scale's contribution: a) and b) the LP variance from the sonic anemometers is compared to the TS variance from the 449 MHz WPR; c) and d) the sonic HP variance is compared to the WPR SW variance; and e) and f) Total Variance$_{\text{sonic}}$ is compared to the sum of the variances from the WPR TS and SW (Figs. 5b, d, and f with $NSPEC = 8$ will be discussed in Sect. 5). With an $R^2$ value of $0.74$, the agreement between TS and LP at $NSPEC = 1$ is strong, with a slope of the best fit line of $0.724$ (Fig. 5a). The largest errors occur for radar TS variances that are significantly higher than the sonic anemometers' LP variance. The average overestimation of the WPR by three (or more) times the sonic anemometers comes mostly from the small variance values, but at the highest values, the agreement is much better (see the departure of the red-dashed best fit line from the black-dashed one-to-one line).

The correlation between the radar SW variances and the HP variance for $NSPEC = 1$ (Fig. 5c), with $R^2 = 0.53$, has a different behavior, with a large over-estimation of small variances, and frequent under-estimations at large variances, as highlighted by the slope of the best fit line much less than 1. At this short time-separation scale, the variance from WPR spectral widths is inaccurate at almost all variance levels. It is also noteworthy that the magnitude of variance is larger overall at the large scale (TS and LP) than the small scale (SW and HP).

The sum of the two portions of the radar's variances is compared to Total Variance$_{\text{sonic}}$ in Fig. 5e. Though dominated in magnitude by the large scales, the spread of values is more condensed than the large-scale values in Fig. 5a, and remains closer to the one-to-one line than the small-scale variances in Fig. 5c. With an $R^2$ value of $0.78$, the agreement is overall better than either of the apportioned contributions. This agreement is very encouraging, showing that it is possible to measure vertical velocity variance with reasonable accuracy from the volume-measurements of the WPRs.

## 5 Spectral Averaging Effects on Variance Measurements

Averaging multiple Doppler spectra in time can reduce the noise level in the radar measurements, and has implications for the scales of turbulence observed in either the spectral width or the time series of vertical velocity. The typical WPR setup optimized for wind measurements (first moment





computations) uses multiple beams pointing in different directions to obtain winds for every 2-5 minutes in order to capture a representative sample of atmospheric motions, while still observing a relatively stationary atmosphere. When analyzing the variance measured by a WPR on two different time scales, it becomes a relevant question of how much averaging should be performed to get

the most accurate measurement for each scale. For example, an optimization of spectral width measurements to be used in turbulence dissipation rates (Hocking, 1985) will call for a different time scale than variances using the time series of resolved vertical velocities from a WPR. Averaging over longer dwells moves more variance contributions into the spectral width, at scales smaller than the dwell time, and out of the time series, increasing the spectral widths, and reducing the contribu-

tion of the variance from the resolved-scale measurements. For a sonic anemometer, averaging over longer time scales simply moves LP variance into the HP variance, until, averaging up to 30 minutes, HP would equal the total variance. However, for a WPR, it is unrealistic for the spectral width of a 30-minute dwell to accurately capture the total variance. It remains to be seen if the radar and sonic anemometers measure the same variances as the information is moved from one set of scales to the

other; the spectral averaging of the WPR and the time series averaging of the sonic anemometers deal with the additional information differently, so the final variances may vary as well. How each scale of WPR observations, as well as the sum of the two, compares to the equivalent variance from the sonic anemometers as the separation time scale lengthens is unknown.

Figure 6 shows the mean absolute error (a), normalized bias (WPR minus sonic divided by sonic,

b) and coefficient of determination, $R^2$ (c), for each set of variances compared in Fig. 5 as a function of the numbers of spectral averages. The correlation between the WPR TS and the sonic anemometer LP variance decreases with longer dwells (more spectral averages), while the bias and MAE increase (MAE more gradually than the normalized bias). The reduction in agreement is visible from Fig. 5a to b, which uses $NSPEC = 8$, indicating that the most accurate measurements of variance from the

WPR time series of vertical velocity are obtained by utilizing the highest temporal resolution data possible, which requires no spectral spectral averaging and short dwells.

On the other hand, the correlation and bias improve between the sonic anemometer HP and the WPR SW variances as more spectral averages are computed. The MAE does increase with longer averages, but the normalized bias's behavior shows that the MAE increase occurs at only larger val-

330 ues of variance, skewing the MAE high, while the normalized behavior shows improvement. The correlation is at its maximum between $NSPEC = 8$ and $NSPEC = 21$, but the MAE increases over that range, so $NSPEC = 8$ is optimal. This optimal number of averages shows improvement in variance at small scales (HP vs. SW), between Figs. 5c and d. This may indicate that the widths observed at short time scales ($NSPEC = 1$, and $\Delta t = 13s$) are mostly dominated by remaining noise,

and are not due to the true atmospheric turbulence. Furthermore, on these short time scales, turbulence has greater spatial variability, so the two instruments, located 600 m apart, may not observe the same value. Over 8 spectral averages, which is equivalent to about 2-minute dwells, the spatial vari-





ability between the two instruments will be reduced. As the averaging time increases, there is also an overall increase in the magnitude of the variances from SW, but there is no apparent decrease in the

340 magnitude of the TS variance, as the energy is moved from one scale to the other. Again, the average overestimation of the WPR SW by three times the sonic HP occurs mostly from the small variance values (the larger difference between the red-dashed best fit line and the black-dashed one-to-one line), but at the highest values, the agreement is much better.

With the improved small-scale SW variance but worsened large-scale TS variances with longer

spectral averaging, it is reasonable that the sum would remain equally correlated with the total sonic variance over all time scales, and this is evident in the correlation (Fig. 6c, purple). While $R^2$ between Total Variance$_{sonic}$ and the sum of the WPR variances remains fairly constant at $0.78 - 0.79$ over all $NSPEC$, the MAE (Fig. 6a) and biases (Fig. 6b) both increase with larger $NSPEC$. The MAE increases at nearly the same rate as the MAE in SW, but the bias increases more slowly than the

bias in TS. The MAE increase in the WPR sum is due to the fact that the magnitude of the SW variance increases with longer dwells (as discussed above), but the TS variance does not decrease to keep the total equal. Since this behavior occurs at all variance levels, the normalized bias increases slower than the bias in TS, which increases drastically with averaging. The main difference between Figs. 5e and f is the slightly larger magnitude of all points, due to the increase in SW values.

With confidence in the agreement between the corresponding sonic anemometer and WPR measurements at 13-s and 2-min scales, and the agreement between the sum of the sonic LP + HP versus Total Variance$_{sonic}$ at 13-s, the agreement between the two sums (sonic LP+HP and WPR TS+SW) was also investigated. The correlation, MAE and bias between the two sums is virtually equal to those of Total Variance$_{sonic}$ vs. WPR TS+SW for all $NSPEC$, indicating the strong correlation be-

tween the sum of the LP and HP and Total Variance$_{sonic}$ that is independent of the separation time scale. The comparison between these with varying $NSPEC$ (using the 449 MHz WPR dwell times) is performed in Fig. 7: a) the mean bias as the sum minus the total variance normalized by the total; b) and the coefficient of determination. As expected, the $R^2$ values are close to 1, and the bias is low for all $NSPEC$. As the time scale of separation changes, the variance contributions shift from the

LP portion to the HP portion, and their sum overestimates the total variance slightly. This positive bias in the sum comes from the remaining low-frequency trends in the HP variance, which decrease with longer averages. Overall, however, the agreement between the Total Variance$_{sonic}$ and the sum of HP and LP is quite good, confirming the accuracy of Eq. 10 for all $NSPEC$.

The collection of comparisons in Figs. 6 and 7 shows that the WPR and sonic anemometers do not

respond to changes in the averaging time scale in the same manner. The optimal time scale for the total variance as the sum of WPR variances is the shortest dwell time, with no spectral averaging. The WPR's measurements vary as well; the TS variance correlates best with the sonic anemometers' LP variance at short time scales, while the WPR's SW variance correlates best with the sonic anemometers' HP at slightly longer, 2-5 minute time scales. Based on these results, if Total Variance$_{WPR}$ is the





desired quantity, then no spectral averaging should be performed ($NSPEC = 1$), gaining the high-est correlation with the lowest biases. However, if variance from the spectral widths is the desired quantity (for calculation of dissipation rates, for example), then the highest correlation and lowest biases occur at $NSPEC = 5 - 10$. For further analysis herein, we use $NSPEC = 8$.

## 6   Effect of Stability

Since the time scales of turbulence are impacted by convection in the planetary boundary layer, an analysis was completed to understand if the time scale at which the WPRs measure the most ac-curate resolved and unresolved variances is affected by the stability of the atmosphere. Data were separated into daytime (convective) and nighttime (stable) sets, and the same comparisons were made. Figure 8 shows the a) MAE, b) normalized bias (sonic minus WPR divided by sonic), and

c) coefficient of determination, $R^2$, for each pair of variances in the daytime and nighttime, with increasing $NSPEC$. The overall result is that the daytime, convective variance (solid lines) is better measured by the WPRs in all methods, following the same behavior as the entire dataset in the pre-ceding sections. In the nighttime stable boundary layer, when turbulence is suppressed, the WPR is not as accurate (dashed lines). The magnitudes of the MAE are smaller at night because the overall

amplitude of the variance is smaller, but the normalized bias shows the larger error at night. Even at night, we see the correlation decrease with increasing $NSPEC$ for the TS vs. LP variances, but increase between WPR SW and sonic HP. In both night and day, the sum of WPR stays equally correlated at larger $NSPEC$, but with increasing MAE, again supporting the use $NSPEC = 1$ for Total Variance$_{\text{WPR}}$. Figure 9 shows the daytime (left column) and nighttime (right column) scatter-

plots of variances, using the optimum $NSPEC$ for each method ($NSPEC = 1$ for TS vs. LP and TS+SW vs. Total Variance$_{\text{sonic}}$, and $NSPEC = 8$ for SW vs. HP). Beside the increased number of observations of small variances at night, the scatter is increased at both large and small scales, and ultimately the sum as well. The low variances that occur at night are inherently more difficult for the WPR to measure, since the remaining noise in the Doppler spectrum can dominate the small

turbulent contributions to the measured spectral widths.

## 7   Results from the 915 MHz WPR

The 915 MHz WPR was situated within 20 m of the 449 MHz WPR for the extent of XPIA, so it pro-vides another opportunity to test the ability of WPR systems to calculate vertical velocity variance. The 449 and 915 MHz WPRs were set up to have very similar spectral and temporal resolution, but

have different parameter sets that produce these desired values (see Table 1). The filtering methods and moments' calculation methods are independent of the WPR parameters, but the number of spec-tral averages, which impacts the SNR and depends on the exact temporal resolution of each WPR system, must be tested for the 915 MHz WPR independently from the 449 MHz results. Using the





same post-processing techniques, the a) MAE, b) bias, and c) coefficient of determination between
variance from the WPR TS and SW and sonic LP and HP variances are shown in Fig. 10, with vary-
ing $NSPEC$. Though the overall error is higher, and correlation is lower due to the inherently nois-
ier 915 MHz system, the behavior is consistent with the results from the 449 MHz WPR. The WPR
TS and sonic anemometer LP become less correlated and more biased with longer dwells, due to the
smaller number of velocity observations that contribute to each variance measurement, but with rel-
atively constant MAE. The correlation between the WPR SW and sonic anemometer HP increases
with longer dwells, but also has increasing MAE. However, the normalized bias is constant with
increasing $NSPEC$ (Fig. 10b). The sum of the WPR TS and SW correlates to Total Variance$_{sonic}$
nearly equally at all time scales as well. The main difference between the 915 MHZ and 449 MHz is
that the variance from TS vs. LP remains better correlated than SW vs. HP up to 5-min dwells. There-
fore, the optimal dwell time for SW variance from the 915 MHz may be longer than the 449 MHz, up
to $NSPEC = 35$, or 10-min dwell time. Figure 11 shows the distributions of variance observations
at each scale (a - d), and Total Variance$_{sonic}$ (e and f), using no spectral averaging ($NSPEC = 1$,
left column), and $NSPEC = 35$ (right column). Again, the improvement in agreement in variance
from WPR SW and sonic anemometer HP can be seen from the left column to the right (c to d), but
a digression is seen in the variance from WPR TS and sonic anemometer LP (a to b). At these longer
time scales, only 3 points contribute to creating the 30-minute variance, so the large scale variance
is not expected to be accurate. The agreement between the WPR sum and Total Variance$_{sonic}$ (e to
f) also increases at $NSPEC = 35$, dominated by the contributions at the small scale in the SW and
HP variances.

**8    Contributions of Measurements to Total Variance**

With two different scales of measurements contributing to the total variance in the atmosphere, the
relative contributions of each can be analyzed. Over the range of variances observed by the 449
MHz radar, the ratio of WPR TS and SW to the sum can illustrate where each scale contributes to
the total variance. Figure 12 shows the ratios of the average observed WPR TS (blue) and SW (red)
to the sum of TS+SW in bins of Total Variance$_{sonic}$. At large variance values, the contribution from
the large scale, TS, variances increases, as the portion from the SW decreases. At smaller values,
however, the contributions remain constant, with more equal portions from TS and SW. The dif-
ference between the solid ($NSPEC = 1$) and dashed ($NSPEC = 8$) lines shows that the fraction
from the SW is larger with longer averages. In fact, the increase leads to a greater contribution to
the summed variance than the TS until the TS begins its increase at larger variances. It isn't until
Total Variance$_{sonic} = 10^{-1}$ m$^2$ s$^{-2}$ that the TS contributes more variance than the SW. This occurs
because more spectral averaging acts to widen the spectral peak. The resolution of the time series
of vertical velocity also decreases with longer dwell times, and the TS variance thus decreases as





the SW variance increases. In the full energy spectrum, the variance is being transferred from the large scale portion to the small scale portion. However, Fig. 5 shows that the SW variance grows more (panels c to d) than the TS variance decreases (panels a to b) with longer averaging, causing an overall increase in the total or summed variance (panels e to f), and overall higher bias in the summed variance (Fig. 6b).

Having assessed the correlations with the *in situ* observations from the sonic anemometers on the 300-m tower, shown in the figures above, full vertical profiles of vertical velocity variance can now be observed by the two WPR systems. As seen in Figs. 13 and 14, the 449 MHz WPR can nearly continuously measure the variance up to 2 km, and the 915 MHz often measures to 1 km or higher. Variance levels as high as 10 $m^2$ $s^{-2}$ near the surface, and down to $10^{-4}$ $m^2$ $s^{-2}$ aloft are observed by both WPRs. Throughout the days shown, the growth and decay of the boundary layer is visible in increasing variance levels in diurnal cycles. The 499 MHz has a narrow-enough beam that the broadening term does not surpass the measured widths, but the 915 MHz WPR's wider beams require a large broadening term to be removed, often larger than the observed spectral width, and thus small variance values are generally not measured at heights above the boundary layer. As the daytime boundary layer grows, however, the measurement height of the 915 MHz profiler increases, as the convection generates stronger velocities, and larger widths become more decipherable despite the large beam-broadening term for that WPR. With observations every 25 m in the vertical, both WPR systems provide highly-resolved profiles of vertical velocity variance within the PBL.

Profiles created using the optimal settings for the different variances show the relative contributions from each, supporting the results of Fig. 12. In the left columns of Figs. 13 and 14 with no spectral averages, the magnitude of the SW variance is much less than that of the TS variance, and in the right columns, with longer time separations, the magnitude of the SW variance is larger. For observations of the variance from the time series of WPR vertical velocity alone, Figs. 13a and 14a are optimal; for variance from WPR spectral widths alone, Figs. 13d and 14d are optimal, and for the total variance, using the sum of TS and SW, Figs. 13e and 14e are optimal.

## 9 Conclusions

With the goal of improving methods of measuring vertical velocity variance from wind profiling radars, two WPRs were run alongside the 300-m BAO tower with 6 heights of sonic anemometers for two months of the XPIA field campaign. The WPRs were set-up with high $NFFT$ and low $NSPEC$ to optimize both the temporal and spectral resolution, allowing measurement of the highest frequencies possible in the energy spectrum, and also allowing flexibility in post-processing through spectral averaging. The spectral resolution of the obtained Doppler spectra was also set to be much higher than in usual operations, in order to get very accurate spectral widths, and to capture the smallest variances possible. Using the *in situ* observations of vertical velocity variance



from sonic anemometers mounted on the BAO tower, comparisons were made between variances
obtained from the WPRs' vertical velocity time series at large scales, and from the spectral widths
of the Doppler spectra at small scales. After filtering the sonic anemometer data to match the time
scales that the WPR measures, the sum of the sonic LP and HP variances matched Total Variance$_{\text{sonic}}$,
with $R^2 = 0.97$. The LP variance from the sonic anemometers showed good agreement with the TS
variance of the vertical velocity from the WPR with no spectral averaging ($R^2 = 0.74$), while av-
eraging 8 spectra proved to be the most accurate for comparisons of HP variance from the sonic
anemometers and WPR spectral widths (SW), at $R^2 = 0.79$. With confidence in each of these com-
parisons, the sum of the variances from the WPR time series and spectral widths was compared
to Total Variance$_{\text{sonic}}$, showing good agreement, with $R^2 = 0.78 - 0.79$ for all $NSPEC$, and only
slightly increasing in MAE and bias with longer time scales. Depending on the application of the
variance from WPRs, spectral averaging may be desired. For the usage of spectral widths for dis-
sipation rates, for example, longer dwells are optimal, showing the highest correlation, even above
the total variance. For only the large-scale, resolved variance, or the total variance as the sum of the
TS and SW, higher temporal resolution with $NSPEC = 1$ is optimal. Results from the 915 MHz
WPR showed equivalent time scales for the optimal agreement between variances. Further division
of the observations into daytime (convective) and nighttime (stable) boundary layers showed that the
449 MHz WPR has better agreement during the day, when turbulence levels are higher, and noise
contributes less to the Doppler spectra.

With these results, wind profiling radars have been shown to reasonably accurately measure ver-
tical velocity variance over the full range of turbulence scales and magnitudes observed by sonic
anemometers. This allows profiles to be collected with these systems through the PBL without being
limited to the locations of the *in situ* observations. The 449 MHz system observes reliable vertical
velocity variance profiles up to 2 km in the set-up used in XPIA, and the 915 MHz WPR measures
consistently up to 1 km. With the ability to observe profiles of variance throughout the planetary
boundary layer from WPRs, progress can be made in many areas including improving PBL param-
eterizations in numerical weather prediction models.

**Appendix A: Discussion of Noise Contributions in Variance Measurements**

In observations of turbulence, the inherent fluctuations and noise that an instrument introduces to
the true measurements must be accounted for. Even in perfectly laminar flow, instrument noise
would result in non-zero variance observations, whether due to the limited accuracy of the mea-
surements or assumptions made to extract velocity from other raw data, as in the case of WPRs.
The removal of the noise contribution to turbulence observations is completed in many different
ways, depending on the instrument type and its level of accuracy. For example, since the noise in
measurements is uncorrelated from turbulence, Thomson et al. (2010) determined that the Doppler



noise variance, $n^2$, from oceanic acoustic Doppler current profilers and velocimeters can simply be subtracted from the observed variance, $\overline{u'^2}$, to obtain the true variance used in calculating turbulence intensity, $I = \frac{\sqrt{\overline{u'^2} - n^2}}{\overline{u}}$. Spectral methods of estimating velocity variance from the Fourier transform of a velocity times series allows the separation of turbulence and noise through subtraction of the random signal from the power density spectrum (Moyal, 1952). When calculating variance from spectral density curves using spatially-averaged measurements (like sonic anemometers and WPRs), corrections must also be applied to account for path-averaging as well as inaccuracies in using the assumption of Taylor's hypothesis across the measurement volume (Kaimal et al., 1968; Wyngaard and Clifford, 1977).

In the current study, the noise contributions to the variance measured by each instrument must be addressed. In the case of the high-frequency point measurement of the sonic anemometers, the manufacturer-prescribed noise level is $n = 0.1 \; cm \; s^{-1}$, which can be 3 orders of magnitude less than the fluctuations in velocity due to turbulence, so $n^2$ is typically negligible. For the WPR, however, there does not exist an inherent $n$, but rather each dwell has an independent noise level, observed in the signal-to-noise ratio, SNR.

Though the effects of beam-broadening and shear-broadening are removed from the WPR spectral width, there is no equivalent method of removal of noise from variance measurements calculated from the time series of velocities, nor any adjustment for errors in spectral widths due to noise. However, expanding upon the work of Riddle et al. (2012) on the minimum threshold of usability for WPRs based on SNR, the accuracy of spectral width measurements can be determined. Riddle et al. (2012) determined the lowest possible SNR needed to recognize a signal in the spectrum, and adopting his method can identify the true spectral width using an additional SNR, $PR$, above the base level needed. To begin, we assume that the true signal, as a function of velocity, $S(v)$, has a Gaussian distribution with mean velocity, $V_0$, and variance, $\sigma^2$:

$$S(v) = \frac{P_0}{\sigma\sqrt{2\pi}} e^{-\frac{(v-V_0)^2}{2\sigma^2}} \tag{A.1}$$

The moments are defined as Eqs. 2-4, integrating symmetrically based on the velocity at which the noise level is reached, $B$. Integrating Eq. A.1 from $V_0 - B$ to $V_0 + B$ (in Eq. 4) produces the estimator of the width, $W_{obs}^2$:

$$W_{obs}^2 = \frac{\int_{V_0-B}^{V_0+B} (v-V_0)^2 S(v) dv}{\int_{V_0-B}^{V_0+B} S(v) dv} \tag{A.2}$$

$$= \sigma^2 - \sqrt{\frac{2}{\pi}} \sigma B \frac{e^{-\frac{B^2}{2\sigma^2}}}{erf\left(\frac{B}{\sqrt{2}\sigma}\right)}. \tag{A.3}$$

The value of $W_{obs}^2$ will be the most accurate measure of $\sigma^2$ when the SNR is high, since $B$ will be large. The fractional error in the width, $F_{W^2}$, is thus

$$F_{W^2} = 100 * \frac{W_{obs}^2 - \sigma^2}{\sigma^2} = -100 * \left[\sqrt{\frac{2}{\pi}} \frac{B}{\sigma} \frac{e^{-\frac{B^2}{2\sigma^2}}}{erf\left(\frac{1}{\sqrt{2}}\frac{B}{\sigma}\right)}\right] \tag{A.4}$$



Again, with a larger SNR and $B$, the fractional error will be smaller. As seen in Fig. 15, for a fractional error in variance of less than 5%, $B/\sigma$ must be larger than 2.76, or $B/\sigma > 2.45$ for fractional error of 10%.

To relate this value to SNR, we use the ratio of power at the peak of the signal and the power at the integration limits (noise level).

$$\frac{S(V_0 + B)}{S(V_0)} = \left(\frac{P_0}{\sigma\sqrt{2\pi}}\right)\left(\frac{P_0}{\sigma\sqrt{2\pi}}e^{-\frac{B^2}{2\sigma^2}}\right)^{-1} = e^{-\frac{B^2}{2\sigma^2}} \tag{A.5}$$

Using this ratio, the relationship can be established between the observed power and the signal at the integration limits, which has units of dB:

$$PR = 10\log_{10}\left[\frac{P_{obs}}{S(V_0 + B)} = \sigma\sqrt{2\pi}e^{\frac{B^2}{2\sigma^2}}erf\left(\frac{B}{\sqrt{2}\sigma}\right)\right]. \tag{A.6}$$

The $PR$ (power ratio), in dB units, is the SNR above the base level needed to identify the signal (peak). This value is added to the SNR threshold from Riddle et al. (2012) to define the limit of detectability of the spectral width based on SNR:

$$SNR\,\min_W = 10\log_{10}\left[\frac{PR*25\sqrt{NSPEC - 2.3125 + \frac{170}{NFFT}}}{NSPEC*NFFT}\right]. \tag{A.7}$$

The use of the fractional error and this ratio can either provide a level of accuracy for each dwell, based on its SNR, or provide a threshold, given a pre-defined level of accuracy. For example, by first defining a fractional error of 10% a value of $B/\sigma$ great than 2.76 and $PR$ of 20.51 dB is required, which, for the 449 MHz at $NSPEC = 8$ and $NFFT = 16384$, equates to a minimum SNR of -20.61 dB. This requirement is always satisfied and therefore, this system is not contaminated by noise enough to prevent to identification of second moments, within 10% accuracy. For the 915
565    MHz, at 10% accuracy, a SNR threshold of -11.56 dB is required. Even with the lower SNRs in that system, this stricter threshold does not reject any more points than the base threshold in Eq. 6. Though it holds in theory, the non-Gaussian basic behavior of the WPR spectra does not allow for this threshold theory to apply to the degree of detail it requires. Further experimentation with the
thresholding method, especially for WPRs set up with such high spectral resolution, is needed for application to these turbulence measurements.

*Author contributions.* K. McCaffrey completed the primary analysis with the aid of L. Bianco, J. Wilczak. P. Johnston contributed in radar setups and post-processing. K. McCaffrey prepared the manuscript with contributions from all co-authors.

*Acknowledgements.* Thanks are due to Timothy Coleman for his role in data acquisition, and to Dave Carter for his help with data processing in POPN4, and to Chris Fairall for many conversations and much wisdom. KM was funded by the NRC Research Associateship Postdoctoral Fellowship. The XPIA field program was funded under the US Department of Energy's Atmospheres to Electrons (A2e) program and by NOAA/ESRL.



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





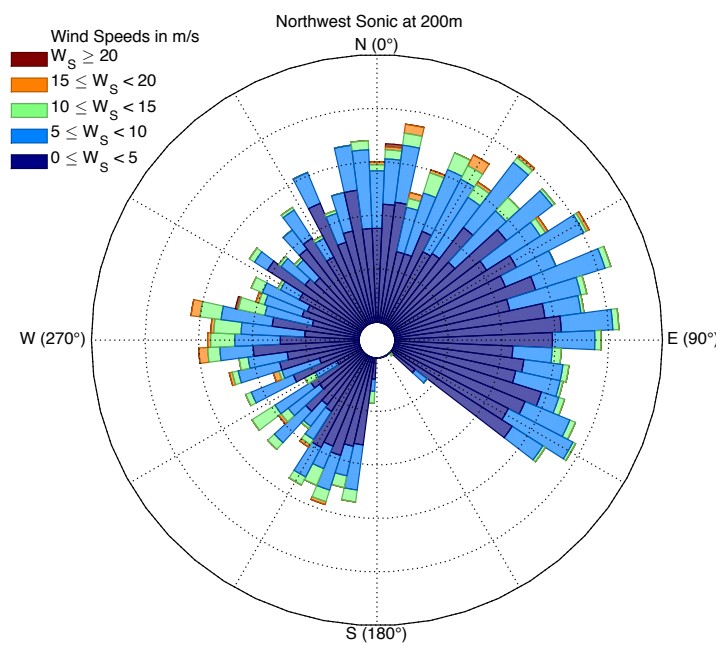

**Figure 1.** Windrose from the 30-minute mean winds measured by the sonic anemometer on the northwest boom at $200m$ on the BAO tower. Waked measurements have been removed and appear as a gap in observations around $154°$.

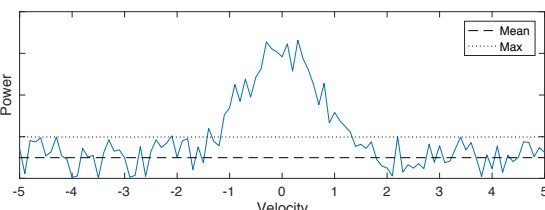

**Figure 2.** Theoretical Gaussian Doppler spectrum with added random noise, with the mean (dashed line) and maximum (dotted line) noise levels.





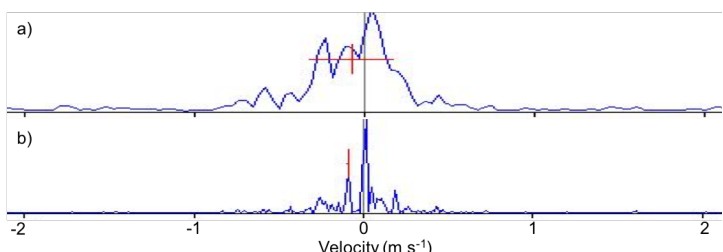

**Figure 3.** Doppler spectra collected from the 499 MHz WPR during the XPIA field campaign, with typical spectral resolution (a) and higher spectral resolution (b), accomplished through computing fewer spectral averages on the same dwell. The vertical red lines denote the first moments (mean velocity) and the horizontal red lines denote the spectral widths, using the standard peak processing method.

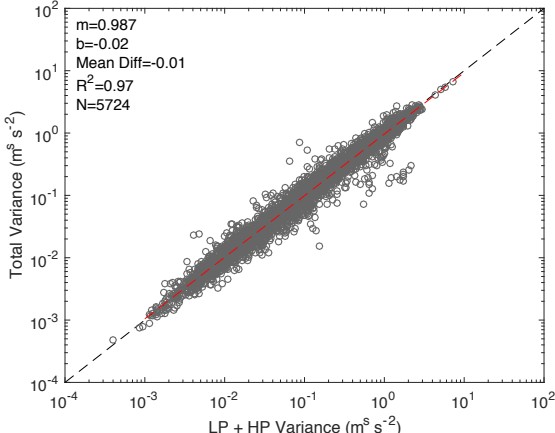

**Figure 4.** Scatter plot of the total sonic anemometer variance versus the sum of low-passed (LP) and high-passed (HP) variances from sonic anemometers, with the time-separation interval set to the 449 MHz, un-averaged ($NSPEC = 1$) dwell time of 13 s. The black dashed line is the one-to-one line. Data from all six heights of sonic anemometers, from the start of each instrument's measurements to 30 April 2015, are included. Also shown are the slope (m) and intercept (b) of the best fit line (red dashed line), as well as the mean difference and coefficient of determination and the number of points plotted.




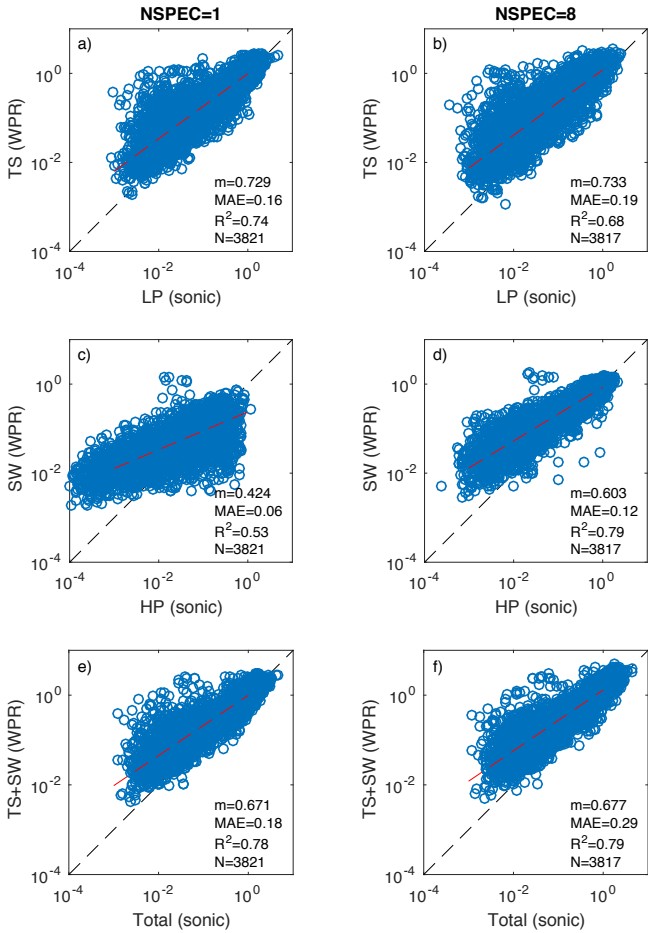

**Figure 5.** Scatter plots of 30-minute vertical velocity variance between the sonic anemometers and the 449 MHz WPR at overlapped heights of 150, 200, 250, and 300-m, for the two months of radar measurements: a) and b) low-passed variance from sonic anemometers (LP) versus WPR time series of vertical velocity (TS); c) and d) high-passed variance from sonic anemometers (HP) versus variance from WPR spectral widths (SW); e) and f) total variance from sonic anemometers versus the sum of TS and SW from the WPR. In panels a), c) and e), no averaging was performed on the WPR spectra, producing a dwell time of 13 s, and in panels b), d), and f) $NSPEC = 8$, generating a dwell time of approximately 2 minutes. The slopes of the best fit lines (red dashed lines), mean absolute errors, $R^2$ values, and number of points, N, are shown for each plot.




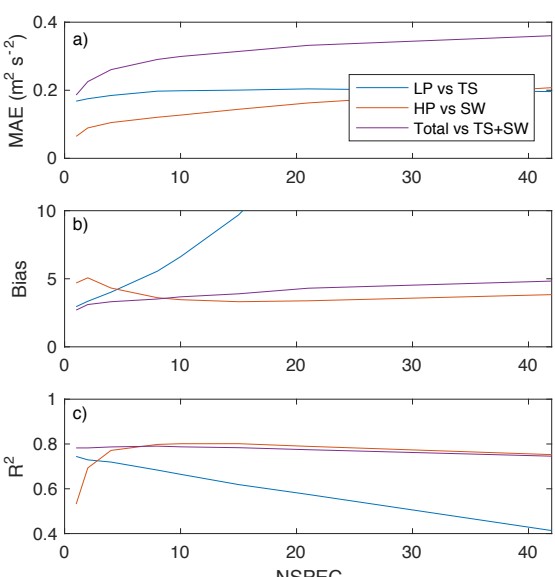

**Figure 6.** a) Mean absolute error, b) normalized bias (WPR minus sonic, normalized by sonic) and c) correlation of determination between sets of variance measurements: low-passed variance from sonic anemometers (LP) versus WPR time series of vertical velocity (TS), blue; high-passed variance from sonic anemometers (HP) versus variance from WPR spectral widths (SW), red; total variance from sonic anemometers versus the sum of TS and SW from the WPR, purple. Data from all four overlapping heights of the 449 MHz WPR and the sonic anemometers and all data from 1 March to 30 April 2015 are included.

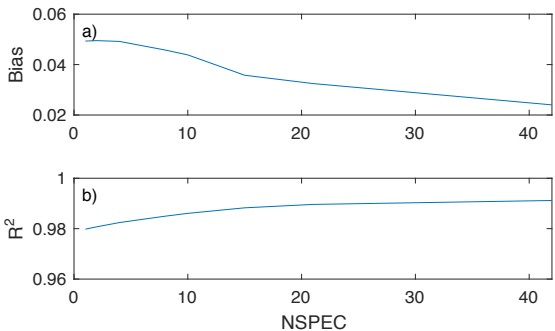

**Figure 7.** a) Normalized bias (sum minus total, normalized by the total), and b) coefficient of determination between Total Variance$_{\text{sonic}}$ versus the sum of low-passed and high-passed variances from sonic anemometers with the time scale determined by the 449 MHz WPR under differing numbers of spectral averages ($NSPEC$). Data from all six heights and all dates of sonic anemometer measurements are included.





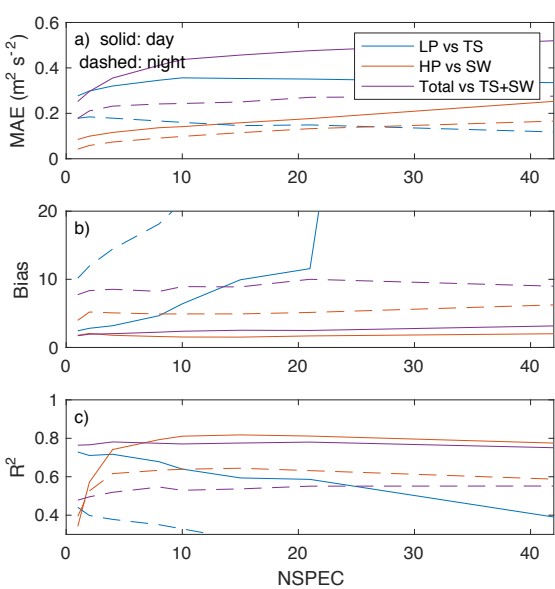

**Figure 8.** Same as in Fig. 6 but separated by daytime (solid lines) and night time (dashed lines).

| Radar freq (MHz) | 449 | 915 |
|---|---|---|
| $IPP(\mu s)$ | 33 | 45 |
| Pulse Width (ns) | 700 | 417 |
| $NCOH$ | 24 | 182 |
| $NSPEC$ | 1 | 1 |
| $NFFT$ | 16384 | 2048 |
| First gate height (m) | 154 | 76 |
| # Range gates | 80 | 72 |
| Range gate height (m) | 26 | 25 |
| $\Delta t$ (s) | 12.98 | 16.77 |
| Spectral Resolution (m s$^{-1}$) | 0.025 | 0.01 |

**Table 1.** Radar parameters for the 449 MHz and 915 MHz wind profiling radars, running in "turbulence mode"
for minutes $25 - 55$ of each hour during XPIA from 1 March to 30 April 2015.





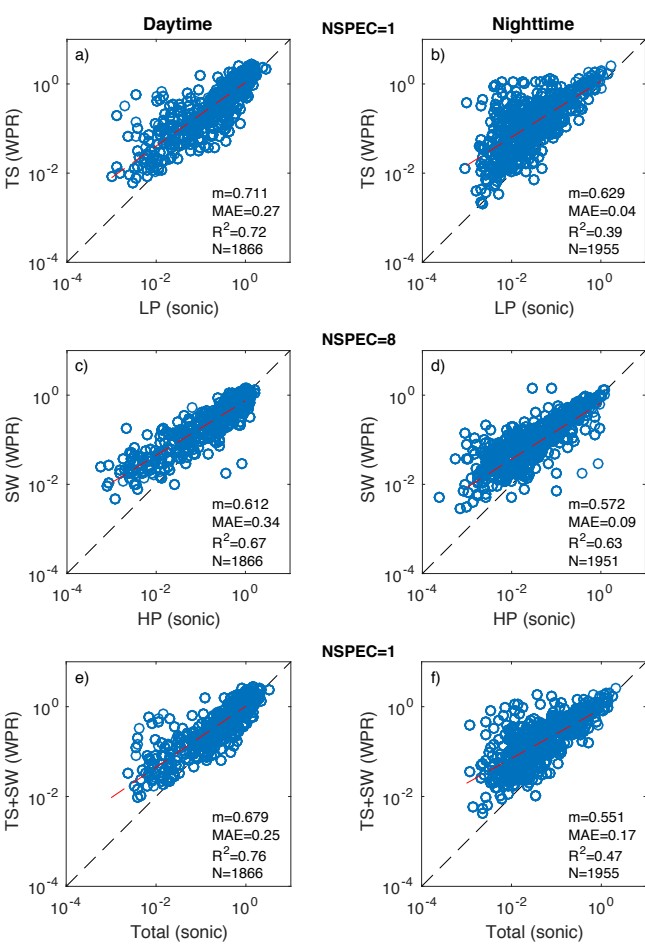

**Figure 9.** Same as in Fig. 5 but separated by daytime (a, c, e) and night time (b, d, f), with the respective $NSPEC$s shown.





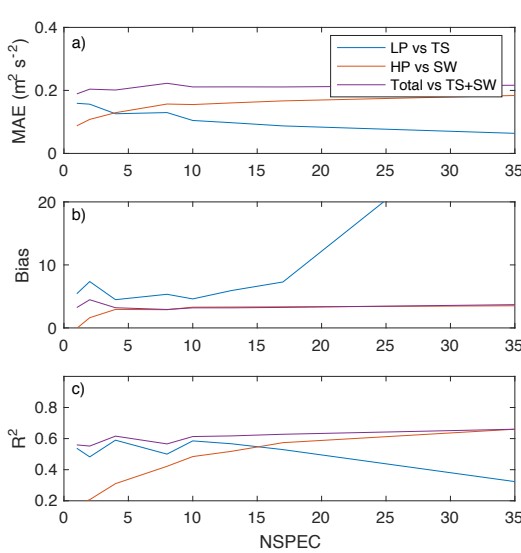

**Figure 10.** Same as Fig. 6, but for the 915 MHz WPR. Note different vertical axis axis on panel b). All six

heights are overlapping, and therefore used in this figure.





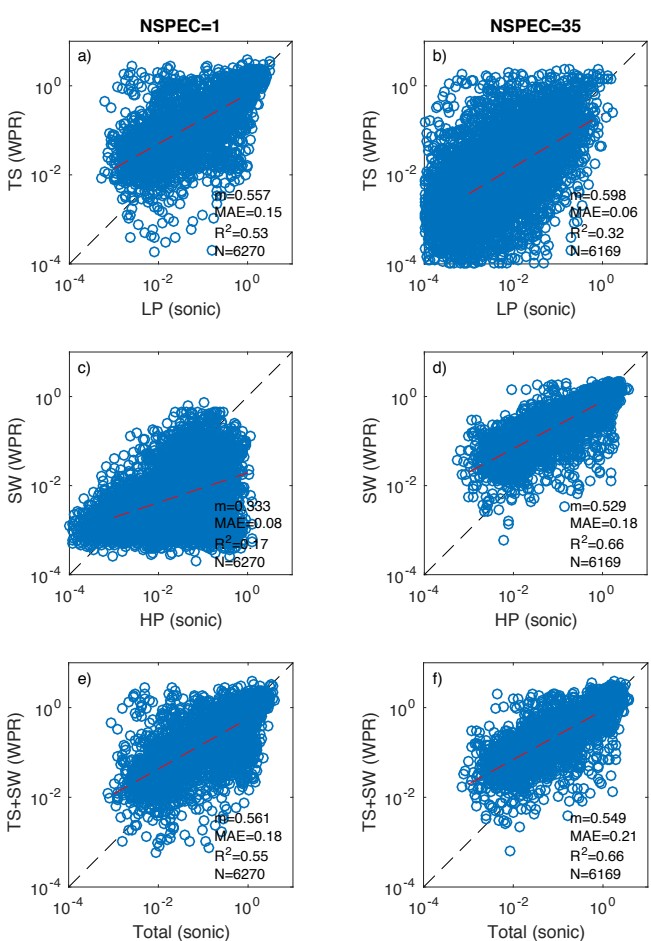

**Figure 11.** Same as Fig. 5, but for the 915 MHz WPR, with $NSPEC = 1$ on the left column, and $NSPEC = 35$ on the right. Data from all six overlapping heights and all available days are included.




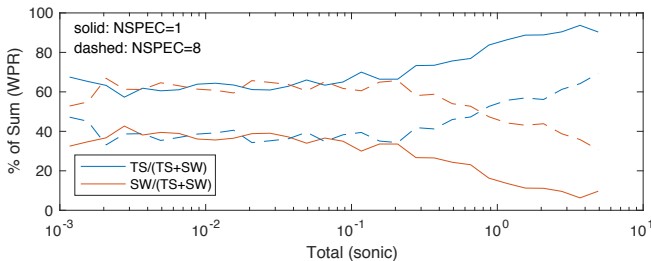

**Figure 12.** Mean percent contribution of the WPR time series (TS; blue) and spectral width (SW; red) variances to the sum of the TS and SW variances binned by the total variance of the sonic anemometers. Solid lines use $NSPEC = 1$, and dashed lines use $NSPEC = 8$, from the 449 MHz WPR at four overlapping heights.

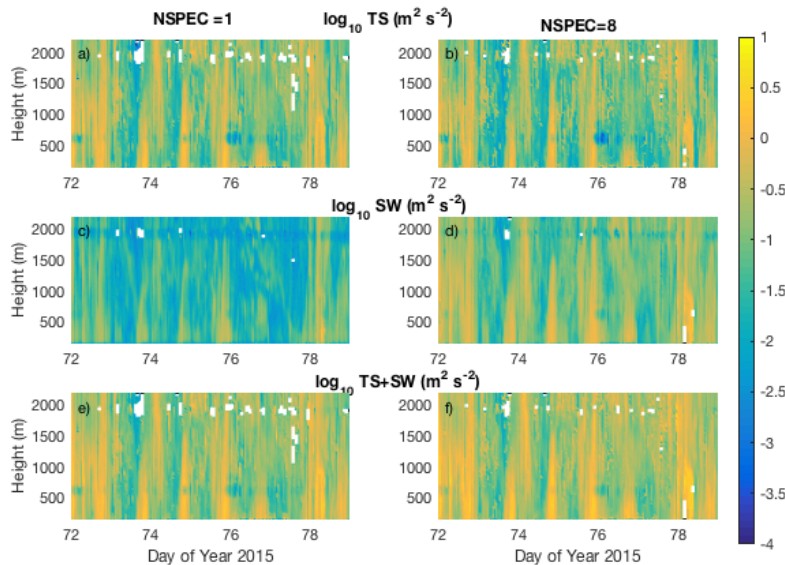

**Figure 13.** Time-height cross-sections of: a) and b) time series vertical velocity variance; c) and d) spectral width variance; and e) and f) total variance as measured by the 449 MHz WPR at the BAO, using $NSPEC = 1$ (a, c, e) and $NSPEC = 8$ (b, d, f), from 13 to 20 March 2015.





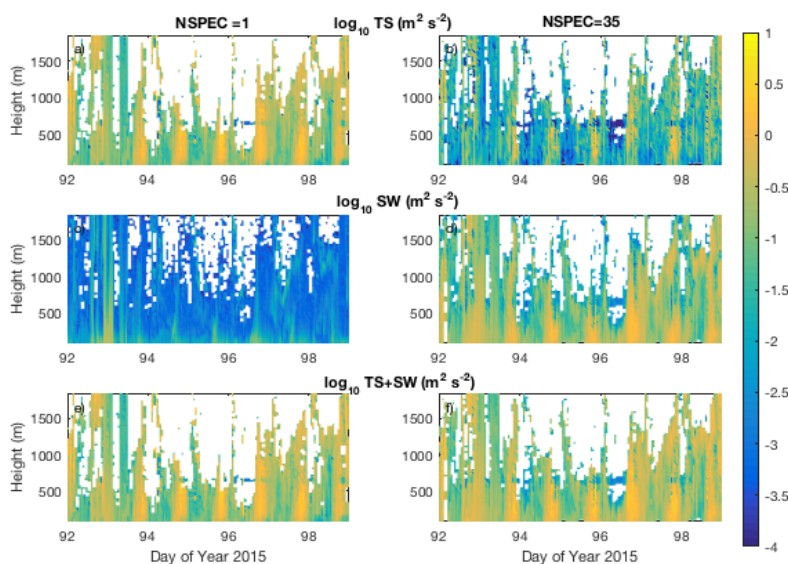

**Figure 14.** Same as Fig. 13, but for the 915 MHz WPR, using $NSPEC = 1$ (a, c, e) and $NSPEC = 35$ (b, d, f).

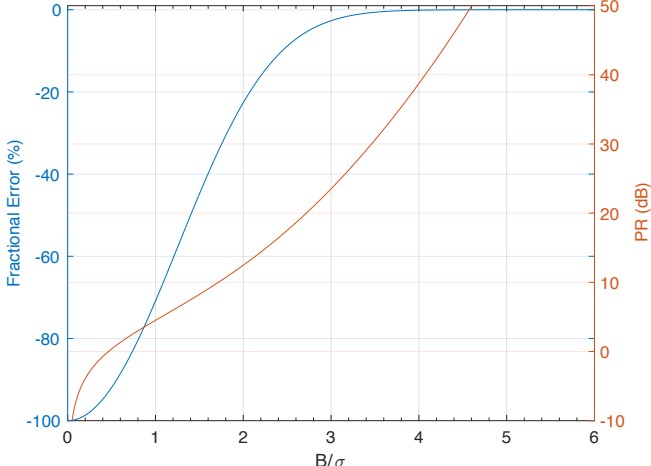

**Figure 15.** Left blue axis: Fractional error of variance from Eq. A.4 as a function of $B/\sigma$. Right red axis: Ratio of observed power to power at noise level integration limits, $PR$ from Eq. A.6, as a function of $B/\sigma$.