# Peer review of "Vertical Velocity Variance Measurements from Wind Profiling Radars"

_Atmospheric Measurement Techniques, 2016_

## Referee Comment (RC1) · S. Jacoby-Koaly (Referee) · 1 Dec 2016

General comments 1) This article presents a solid and well organised study of the measurement of large-scale and small-scale vertical velocity variances by wind profiling radars (WPR) during XPIA campaign by statistical comparison with in situ sonic anemometers measurements. The final purpose is clearly indicated: to identify the best WPR configuration and post-processing methods to measure accurate variances at different scales. 2) The title not adequately describes an important part of the work devoted to the analysis of sonic anemometers data. Authors should amend the title to include reference to sonic anemometers. 3) The results are properly illustrated in sections 4 to 8 but some figure descriptions in sections 4 and 5 are too long, detailed or redundant, which makes the text more cumbersome. Authors need to find a way to better synthetize the comparative results (maybe using tables). 4) Despite some

improvements suggested above, I recommend publication of this paper after minor revision.

Specific comments Lines 32-33: Better rephrase this point in the text. Lines 37-39, 44-45: More references are required in the introductory part to support some statements. Lines 167-168: Add a few words here to better explain your choice. Line 203, Equation 8: Define wr', R0, Theta. Did authors try to get statistical comparative results at discrete levels? In other words, do all height levels follow the same trend?

Technical corrections Line 280: 0.729 instead of 0.724

---

## Referee Comment (RC2) · S. Emeis (Referee) · 16 Dec 2016

This is an important study which gives confidence in remotely measured vertical velocity variances. The study is well written and should be published more or less as is.

Maybe, the Conclusions could be formulated a bit better. The Conclusions should not just repeat the most important numbers from the results section (which already have been highlighted in the abstract as well). The authors should try to formulate the main results in new words, maybe in form of a few bullet points. The conclusions should illustrate the impact of the outcome of the presented research.

Especially, the last paragraph of the Conclusions could be extended. Presently, it just reports the well-known fact that the range of the 449 MHz wind profiler is larger than

the one of the 915 MHz instrument. The results section (see lines 418ff) gives more information on the different abilities of the two wind profilers. The results from lines 418ff should be commented on in the Conclusions as well.

---

## Author Comment (AC1) · 22 Dec 2016

We thank both referees for their very thorough and insightful comments on our manuscript. We have addressed the general, specific, and technical comments, and believe that the changes we've made are a great improvement to the clarity of communicating the work that we have done. (1) Referee's Comments (2) Author's Response (3) Change to Manuscript

We also have attached the revised manuscript as a supplement, with revisions denoted in blue text.
General comments This article presents a solid and well organized study of the measurement of large-scale and small-scale vertical velocity variances by wind profiling radars (WPR) during XPIA campaign by statistical comparison with in situ sonic anemometers measurements. The final purpose is clearly indicated: to identify the best WPR configuration and post-processing methods to measure accurate variances at different scales. (1) The title not adequately describes an important part of the work devoted to the analysis of sonic anemometers data. Authors should amend the title to include reference to sonic anemometers. (2) We've changed it, as requested: (3) Title: "A Comparison of Vertical Velocity Variance Measurements from Wind Profiling Radars and Sonic Anemometers" (1) The results are properly illustrated in sections 4 to 8 but some figure descriptions in sections 4 and 5 are too long, detailed or redundant, which makes the text more cumbersome. Authors need to find a way to better synthetize the comparative results (maybe using tables). (2) In several places, we've taken out some confusing sentences, and re-phrased things. (3) See blue text throughout sections 4-8 Despite some improvements suggested above, I recommend publication of this paper after minor revision.

Specific comments (1) Lines 32-33: Better rephrase this point in the text. (2) This should clarify: (3) line 32: "In the complete energy spectrum, the total variance is made of contributions from the entire range of scales, from large to small. Furthermore, variances are observed at separate scales by different instruments' measurement frequencies and volume sizes." (1) Lines 37-39, 44- 45: More references are required in the introductory part to support some statements. (2) We've reworded and moved the reference at 37-39, since it is all part of Angevine 1994, and cited the references from lines […...] on dissipation rates (3) line 34-40: "In general. . ." line 45: "Hocking. . ." (1) Lines 167-168: Add a few words here to better explain your choice. (2) This should clarify: (3) line 166: "If a non-atmospheric signal produces a high, outlier noise level, the spectral width will be detrimentally narrowed, and therefore, we decided to use the mean noise level with SPP for measurements of Doppler spectral widths because it will give more consistent results." (1) Line 203, Equation 8: Define wr', R0, Theta. (2)

Fixed. (3) line 38: "vertical velocity, w_r," line 204: "(where w'_r is the fluctuation elative to the 30-min mean velocity)," line 216: "where \nu is the half-width to the half-power point in the antenna pattern, \theta is the beam angle from the vertical, R_0 is the lowest range gate, and du/dz is the vertical mean wind shear." (1) Did authors try to get statistical comparative results at discrete levels? In other words, do all height levels follow the same trend? (2) There is a slight difference, but for this study, we focus on the overall effectiveness of the instrument. We've added a short paragraph to this point: (3) line 299: "For the three variables..."

Technical corrections (1) Line 280: 0.729 instead of 0.724 (2) Fixed. (3) line 283

——————————————————-
This is an important study which gives confidence in remotely measured vertical velocity variances. The study is well written and should be published more or less as is. Maybe, the Conclusions could be formulated a bit better. The Conclusions should not just repeat the most important numbers from the results section (which already have been highlighted in the abstract as well). The authors should try to formulate the main results in new words, maybe in form of a few bullet points. The conclusions should illustrate the impact of the outcome of the presented research. Especially, the last paragraph of the Conclusions could be extended. Presently, it just reports the well-known fact that the range of the 449 MHz wind profiler is larger than the one of the 915 MHz instrument. (2) We've added further discussion of the implications of these improved measures, at the end of the last paragraph. (3) line 517: "The evolution..." (1) The results section (see lines 418ff) gives more information on the different abilities of the two wind profilers. The results from lines 418ff should be commented on in the Conclusions as well. (2) There is a new paragraph that summarizes these results. (3)

line 504: "In an analysis..."

Please also note the supplement to this comment:
http://www.atmos-meas-tech-discuss.net/amt-2016-299/amt-2016-299-AC1-
supplement.pdf
* * *
[Figure]

**Supplement:**

Manuscript prepared for Atmos. Meas. Tech.
with version 2015/04/24 7.83 Copernicus papers of the LATEX class copernicus.cls.
Date: 22 December 2016

**A Comparison of Vertical Velocity Variance Measurements from Wind Profiling Radars and Sonic Anemometers**

Katherine McCaffrey[1,2], Laura Bianco[1,2], Paul Johnston[1,2], and James M. Wilczak[2]

[1]University of Colorado, Cooperative Institute for Research in Environmental Sciences at the NOAA Earth System Research Laboratory, Physical Sciences Division, 325 Broadway, Boulder, CO 80305-3337
[2]NOAA Earth System Research Laboratory, Physical Sciences Division, 325 Broadway, Boulder, CO 80305-3337

*Correspondence to:* Katherine McCaffrey (katherine.mccaffrey@noaa.gov)

**Abstract.** Observations of turbulence in the planetary boundary layer are critical for developing and evaluating boundary layer parameterizations in mesoscale numerical weather prediction models. These observations, however, are expensive, and rarely profile the entire boundary layer. Using optimized configurations for 449 MHz and 915 MHz wind profiling radars during the eXperimental

5 Planetary boundary layer Instrumentation Assessment, improvements have been made to the historical methods of measuring vertical velocity variance through the time series of vertical velocity, as well as the Doppler spectral width. Using six heights of sonic anemometers mounted on a 300-m tower, correlations of up to $R^2 = 0.74$ are seen in measurements of the large-scale variances from the radar time series, and $R^2 = 0.79$ in measurements of small-scale variance from radar spectral

10 widths. The total variance, measured as the sum of the small- and large-scales agrees well with sonic anemometers, with $R^2 = 0.79$. Correlation is higher in daytime, convective boundary layers than nighttime, stable conditions when turbulence levels are smaller. With the good agreement with the *in situ* measurements, highly-resolved profiles up to 2 km can be accurately observed from the 449 MHz radar, and 1 km from the 915 MHz radar. This optimized configuration will provide unique

15 observations for the verification and improvement to boundary layer parameterizations in mesoscale models.

**1 Introduction**

Observations of turbulence quantities in the planetary boundary layer (PBL) are crucial for many applications, and in particular, can be extremely informative for developing and evaluating param-

20 eterizations in numerical weather prediction models of the small scales that cannot yet be resolved. However, turbulence measurements are predominantly relegated to high-frequency *in situ* observing instrumentation such as sonic anemometers, limited in their spatial coverage, or are taken by expensive aircraft platforms. Lidar remote sensing instrumentation have demonstrated some potential for measuring profiles of turbulence (Eberhard et al., 1989; Frehlich, 1997; O'Connor et al., 2010), but this technology has more commonly focused on mean wind measurements (Menzies and Hardesty, 1989; Grund et al., 2001; Lundquist et al., 2016). Similarly, wind profiling radars (WPRs) have been shown to have capabilities of measuring turbulence, from information contained in the Doppler spectral width of the vertical velocity (Hocking, 1985; Reid, 1987; Angevine et al., 1994; Nastrom and Eaton, 1997), but the adoption of these techniques into routine use has not occurred because of the lack of precision and inability to measure the smallest turbulence values observed by sonic anemometers.

In the complete energy spectrum, the total variance is made of contributions from the entire range of scales, from large to small. Furthermore, variances are observed at separate scales 
[revised manuscript text omitted]

For the three variables - large scale, small scale, and the total variances - the lowest height of the WPR performs the poorest, with coefficients of determination $R^2 = 0.63$, 0.75, and 0.67, respectively, compared to the overall profile with $R^2 = 0.74$, 0.79, and 0.78. The higher range gates show more constant behavior, indicating that the lowest range gate is an outlier in the profile. If the lowest

measurement of the WPR were removed, the overall statistics would be improved, but since this is a study into the overall effectiveness of the instrument, all heights are analyzed together.

305 ## 5   Spectral Averaging Effects on Variance Measurements

[revised manuscript text omitted]

In an analysis of the contributions that the large and small scales make to the total variance, there are differences that depend on the total variance, and the time scale of separation (set by the number of spectral averages) between scales. Using a larger number of spectral averages move a greater portion of the variance into the small scales, and vice versa. At large total variance levels, the small scales decrease in relative contribution, and the large scales increase. Understanding the scales and levels of variance measured by each method (WPR TS or SW) indicates the best set-up for the WPR, depending on the application.

With these results, wind profiling radars have been shown to reasonably accurately measure vertical velocity variance over the full range of turbulence scales and magnitudes observed by sonic

anemometers. This allows profiles to be collected with these systems through the PBL without being limited to the locations of the *in situ* observations. The 449 MHz system observes reliable vertical velocity variance profiles up to 2 km in the set-up used in XPIA, and the 915 MHz WPR measures consistently up to 1 km. With the ability to observe profiles of variance throughout the planetary boundary layer from WPRs, progress can be made in many areas including improving PBL parameterizations in numerical weather prediction models. The evolution of the PBL can be analyzed in more detail with the use of turbulence profiles from WPRs. Verification of sub-gridscale parameterizations of large-eddy simulations is possible using the small scale variances measured in the WPR spectral widths, while the resolved-scales of the LES can be verified by the WPR variance from the time series of vertical velocities. Furthermore, improved spectral width measurements will allow for more accurate observations of turbulence dissipation rates from WPRs, as performed in McCaffrey et al. (2016b).

[revised manuscript text omitted]